# Asynchronous Partial Gaussian Approximation Detection Algorithm for Uplink-Grouped MIMO-SCMA System

Xue Wang [1], Ninghao Zhou [1] and Jia Hou [1,2,*]

1   School of Electronics & Information, Soochow University, Suzhou 215000, China
2   Yangtze Delta Region Institute (Quzhou), University of Electronic Science and Technology of China, Quzhou 324000, China
*   Correspondence: houjia@suda.edu.cn

**Abstract:** To relieve the impact caused by different arrival times of multiple users in the uplink MIMO-SCMA system, a grouped multiuser transmission model, according to different delays and its corresponding detection scheme, is proposed in this paper. Assuming perfect synchronization in one group, a message-passing algorithm based on serial propagation (SP-MPA) is proposed to reduce the bit error rate (BER), which could transfer the updated information to the next symbol as its initial probability. Furthermore, with a more practical case, the partial Gaussian approximation method (PGA) is designed to decrease the interference resulting from the imperfect synchronization in one group. As the result, the computing complexity of the proposed PGA method could be decreased by at least 20% compared with SP-MPA and the BER could be improved by about 10%.

**Keywords:** sparse code multiple access (SCMA); asynchronous; grouped; multiple input multiple output (MIMO); partial Gaussian approximation (PGA)

## 1. Introduction

With a sharp increase in data and large amounts of connection, communication networks are under unprecedented pressure. The fifth-generation mobile communication system (5G) is expected to support higher requirements and more scenarios [1].

Among all the technologies of 5G, novel multiple-access methods allow multiple users to transmit signals over the same time-frequency resource, which greatly improves the spectral efficiency [2]. Sparse code multiple access (SCMA) is a non-orthogonal multiple-access (NOMA) technology in the code domain that combines symbol mapping and spreading using multi-dimensional codebooks. Benefiting from its sparsity, the detection of multiple users becomes possible in massive machine type of communication (mMTC) [3,4].

For SCMA, a message-passing algorithm (MPA) is thought of as a basic detection method. It is an improvement on the maximum a posteriori (MAP), which iterates the edge probability without exhaustion of all combinations of codewords [5]. Now, more superior methods have been proposed. The authors of [6] proposed a (RRL) detector, which updates the function nodes only within a restricted search region and reduces the targeting points. In [7], a novel fixed low-complexity detector for an uplink SCMA system was proposed based on partial marginalization. Ref. [8] studied an improved sphere decoding for SCMA that achieves the optimal maximum likelihood detector for any arbitrary regular or irregular constellations. In [9], a generalized sphere-coding SCMA was proposed where sorted QR decomposition and Schnorr–Euchner enumeration are applied. Ref. [10] compared MAX-Log-MPA and Log-MPA, which aim to reduce the exponential operations. A deterministic message-passing algorithm in [11] utilized algorithmic simplifications for SCMA decoding, such as domain changing and probability approximation. The authors in [12] proved that serial scheduling-based MPA can reach a faster convergence by updating information with newly updated information in the same iteration.

The methods mentioned above mainly focus on a synchronous SCMA scheme, in which all signals are considered to arrive at the base station at the same time. However, in a real uplink channel, signals come from users in different directions or with different distances, resulting in different delays of transmission. Moreover, a grant scheme produces significant signaling cost, so grant-free transmission is introduced as a competitive alternative in 3rd Generation Partnership Project (3GPP) Release 16 [13]. That means delays of signals are completely random, adding great difficulty for uplink detection. Thus, exploring useful asynchronous detection algorithms is essential.

Some research paid attention to asynchronous detection. Ref. [14] devised a message-passing receiver for the uplink SCMA that uses expectation propagation to tackle the intractable distributions. Ref. [15] estimated a channel based on both received short pilot and data sequences and introduces auxiliary active-state indicators in joint detection. Ref. [16] proposed a blind detection algorithm for detecting user signals without the knowledge of a user codebook. Ref. [17] performed signal detection using deep learning with unknown channel information and system sparsity. Ref. [18] investigated an algorithm based on belief propagation and equal gain combining under asynchronous circumstances and effectively improves the system performance.

It is noted that delays can be various when connections of users are massive, which means the contents of the mixed signal received can be too various to decode in one scheme. Although the algorithms above perform well in an asynchronous system, few of them consider the problem caused by the variety of delays. Thus, making a detection rule is very important to relieve the bad effects brought by interference between symbols.

To further improve system performance, the multiple-input multiple-output (MIMO) technique is used to increase system reliability through diversity gain. Ref. [19] made a selection of receive antennas that minimizes the signal-to-interference ratio of the observed signal during the MPA calculations. Ref. [20] developed low-complexity iterative detection algorithms based on an expectation propagation algorithm. Ref. [21] studied spatial multiplexing (SMX), spatial modulation (SM) and space shift keying (SSK) techniques for the MIMO-SCMA systems. Ref. [22] proposed a partially active message passing based on the Gaussian approximation principle and a sliding window is introduced to determine what user stays active during each detection. The combination of MIMO and SCMA can enhance the performance greatly but it also adds computing complexity at the same time. Ref. [23] proposed a multi-dimensional space-time block code (MSTBC)-aided downlink MIMO-SCMA, which has almost the same complexity but better performance. In [24], a low-complexity codebook design algorithm based on the cross-entropy method and low-complexity MIMO-SCMA codeword detectors based on depth-first and breadth-first tree-search algorithms was proposed. Ref. [25] proposed a new scheme, named spatial modulation sparse code multiple access (SM-SCMA). At the receiver, a low-complexity joint message-passing algorithm is proposed.

Targeting the problems above, this paper constructs a grouped-detection scheme, in which users with similar delays are grouped together for the ease of joint detection. First, we assume that the scheme in the same group is perfect synchronization, which is to say that the delays of users' information in the same group are exactly the same and delays are different between different groups. A message-passing algorithm based on serial propagation (SP-MPA) is proposed and utilized to propagate the updated information to the next symbol as the initial information. This method can effectively cut down the bit error rate because it sufficiently utilizes the same information without recalculation. Then, we consider the imperfect synchronization of users in the same group. In this scheme, performance undergoes a sharp degradation because of the interference of the delayed symbols. Therefore, we propose a partial Gaussian approximation-aided (PGA) algorithm to compensate the error and improve the transmission accuracy when ensuring lower complexity compared to original SP-MPA in the MIMO-SCMA system.

The contents of this paper are outlined as follows. Section 2 represents the asynchronous SCMA system model. Section 3 describes the detailed process of asynchronous

detection using SP-MPA. The imperfect scheme in the same group and the process of PGA are illustrated in Section 4. Section 5 shows the final results of these algorithms, such as bit error rate and complexity. Finally, conclusions are drawn in Section 6.

## 2. System Model

In this paper, the uplink MIMO-SCMA system is constructed with one base station, $K$ frequency resources and $L$ users. Each user is equipped with one antenna and the base station has $R$ antennas $\gamma_r$, $r = 1 \cdots R$. The distance from each user to the base station is $d_l(1 \le l \le L)$. In this system, users are located in various directions and distances and send their information at any time. Users within the coverage of the base station are divided into different groups according to different delays.

### 2.1. Grouping Rule

As shown in Figure 1, different delays of the asynchronous multi-users would lead to various symbol detection cases. It is hard to perform signal sampling and detection algorithms without grouping. Therefore, a grouping rule will help improve the system performance through categorizing similar delays into one group.

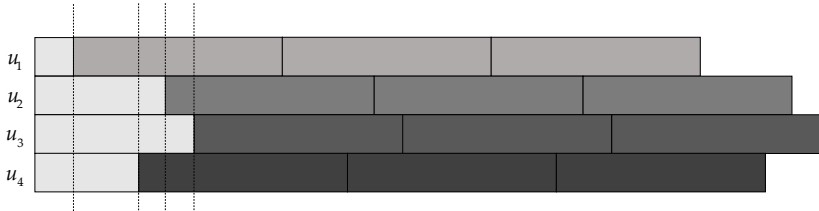

**Figure 1.** Different delays in the detection of asynchronous multi-users.

Assuming that all users are in the same environment but only with different distances to the base station in the system model, we can divide the users with similar distances into one group.

For example, there are two groups in the system model. The users with the distances similar to the $d_{g_1}$ could be grouped as $g_1$ and the users with the distances similar to the $d_{g_2}$ could be grouped as $g_2$. The users in both groups are defined as $u_{g_1}^{(j)}$ and $u_{g_2}^{(j)}(1 \le j \le J)$. The transmitting power of these users is normalized to $P_{g_1}$ and $P_{g_2}$ as the identity. The modulation order is $M$. The delay of two different user groups to the base station is $\tau_{g_1}$ and $\tau_{g_2}$ respectively and the arrows mean the transmitting direction, as Figure 2 illustrates.

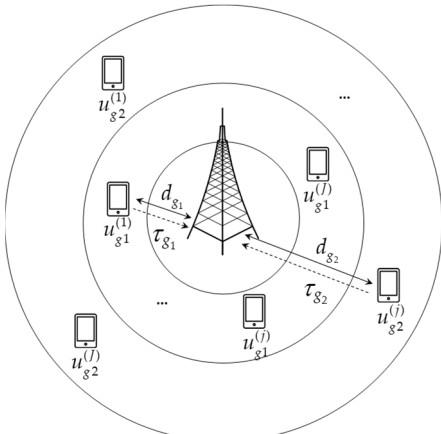

**Figure 2.** Multi-user grouped model.

### 2.2. Codebook

First, the mother codebook for each user group is designed, divided into diversity groups and assigned to users shared on each resource block. Each user maps their bit stream to its own codebook and transmits the mapped symbols to the base station.

The SCMA encoder maps $\log_2(M)$ bits to a $K \times M$ codebook, $\mathbb{B}^{\log_2(M)} \to X$, $X = [x^{(1)}, x^{(2)}, \cdots, x^{(J)}]$. In this model, each user occupies $d_k(d_k < K)$ resource blocks and each resource block accommodates $d_j(d_j < J)$ users.

$$d_j = \frac{J \times d_k}{K} \tag{1}$$

$$J = \binom{K}{d_k} = \frac{K \times (K-1) \times \cdots \times (K - d_k + 1)}{d_k \times (d_k - 1) \times \cdots \times 1} \tag{2}$$

Two mother codebooks, $\mathcal{C}_{g_1}$ and $\mathcal{C}_{g_2}$, are constructed with $\mathcal{C}_{g_1}$ assigned to near users and $\mathcal{C}_{g_2}$ assigned to far users. The radius of $\mathcal{C}_{g_1}$ is bigger than $\mathcal{C}_{g_2}$ and there is a rotated angle between them for distinction. In the uplink system, near users have better channel states, which are easier to be decoded. Thus, $P_{g_1}$ is set bigger than $P_{g_2}$ to obtain a more accurate result because of the joint detection. The mother codebooks are divided into $d_j$ diversities, respectively, to distinct users shared on the same resource block. Figure 3 shows the codeword distribution on the same resource block and users in these two groups both adopt QPSK modulation. The power of signals is normalized and the radius means the transmitting power of users in two groups. Constellation points in each group are $M \times d_j(d_j = 2)$ in Figure 3.

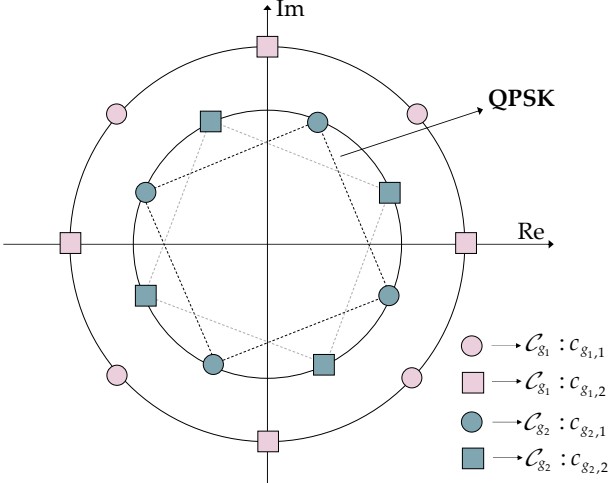

**Figure 3.** Codebooks of multi-user group on the kth resource.

### 2.3. Asynchronous Transmission Model

In the uplink, multiple users transmit data to the base station. Due to the different locations, the channel conditions are different and the fading and loss are also different. Users with similar distances to the base station are divided into one group, so that delays of the users in the same group could be assumed as the same value if the perfect detection scheme is used and the total delay of two different user groups to the base station is $\tau_{g_1}$ and $\tau_{g_2}$, respectively. The symbol '+' means that the noise is additive and '$\sum$' means addition operation, as shown in Figure 4.

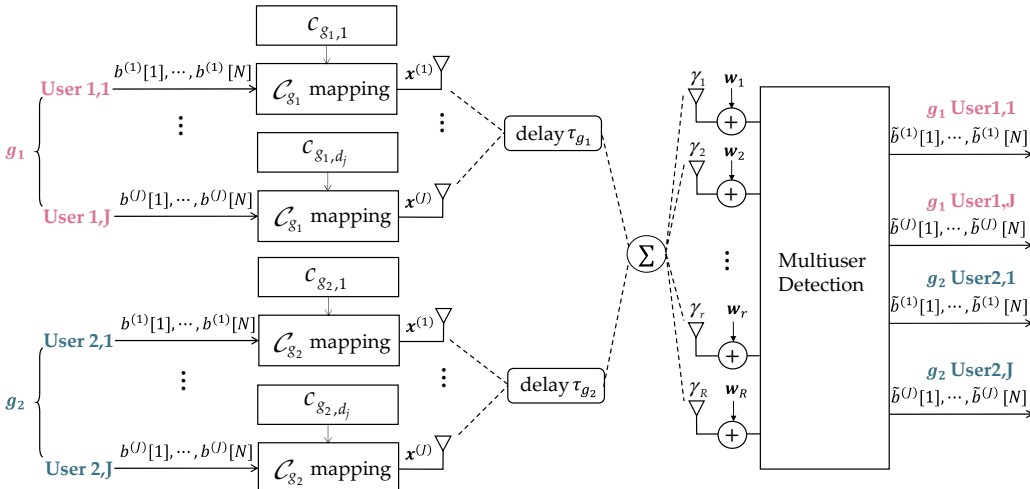

**Figure 4.** Uplink asynchronous MIMO-SCMA multi-user group system.

In the asynchronous scheme, the superimposed signal received at the base station can be expressed as $y = [y_1, y_2, \cdots, y_k, \cdots y_K]^{\mathrm{T}}$, $1 < k < K$. The resource blocks are orthogonal to each other, where OFDM is conducted, so that separate detection of frequency resource blocks can be performed. The superimposed signal on the same resource block is shown as below, where $\mathrm{T}_k$ means the set of users on the $k^{th}$ resource block, $\mathrm{T1}_k = \mathrm{T2}_k = \mathrm{T}_k$. $\tau_{g_1}$ can be written as 0, which is chosen as the reference. However, it is still marked in the following figures in order to be clearer. In addition, due to the similarity of delays in one group, we use rectangles with different colors to represent $\mathrm{T1}_k$ and $\mathrm{T2}_k$. The ratio of $\tau_{g_2}$ and $t_s$ is defined as $\rho$, where $t_s$ means the lasting time of one symbol. Classification discussion of $\rho$ is listed below and Figure 5 shows the transmission situations with different $\rho$.

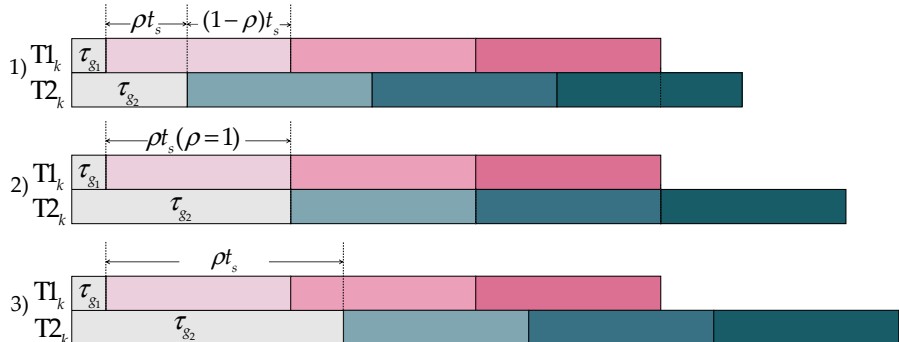

**Figure 5.** Different transmission situations with different values of $\rho$.

1. $0 < \rho < 1$: the delay is within a symbol duration.
2. $\rho = 0, 1, 2, \cdots, N - 1$: the delay is equal to an integer multiple of the symbol duration.
3. $v < \rho < \mu(v = 1, 2, \cdots, N - 1, \mu = v + 1)$: the delay is shifted $v$ symbol durations backward.

As shown in the diagram, situation 2 can be seen as a special case of situation 3, which is categorized into 3. The main difference between 1 and 3 is the first symbol where there is no interference of group 2 to group 1. However, if the first symbol is decoded, detection methods of the left symbols can be the same as 1. Without loss of generality, the following research is only discussed in light of situation 1. Figure 6 shows the complete transmission process.

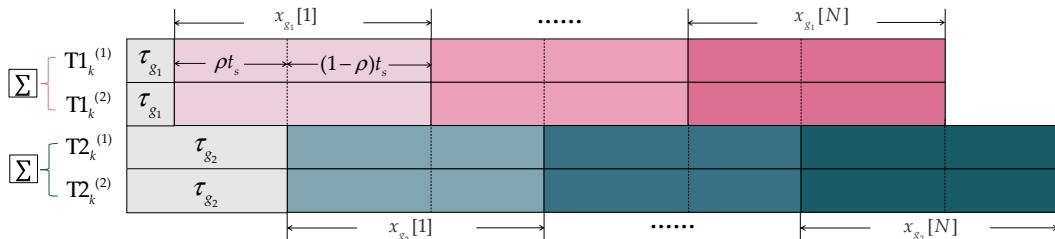

**Figure 6.** Diagram of two-group asynchronous transmission process.

The mathematical expression of the received signal in symbol-related form is represented below and the receiving structure is shown in Figure 7. Because of asynchronization, each symbol not only affects the current symbol but also affects the adjacent symbols [26]. For example, the second symbol of users in group 1 can be affected by the first and second symbol of users in group 2.

$$
\begin{aligned}
y_{k,g}[n] &= \sum_{j=1}^{d_j} h_{k,g}^{(j)}[n] \times x_g^{(j)}[n] + \sum_{g \neq g} \sum_{\psi = 0, \pm 1} \xi_g^{\psi} \\
&\times \sum_{j=1}^{d_j} h_{k,g}^{(j)}[n + \psi] \times x_g^{(j)}[n + \psi] + w_k[n], n = 1, \cdots N
\end{aligned}
\tag{3}
$$

$$
\xi_{g1}^{\psi} = \begin{cases} 1 - \rho & , \psi = 0 \\ \rho & , \psi = -1 \\ 0 & , \psi = 1 \end{cases}
\tag{4}
$$

$$
\xi_{g2}^{\psi} = \begin{cases} 1 - \rho & , \psi = 0 \\ 0 & , \psi = -1 \\ \rho & , \psi = 1 \end{cases}
\tag{5}
$$

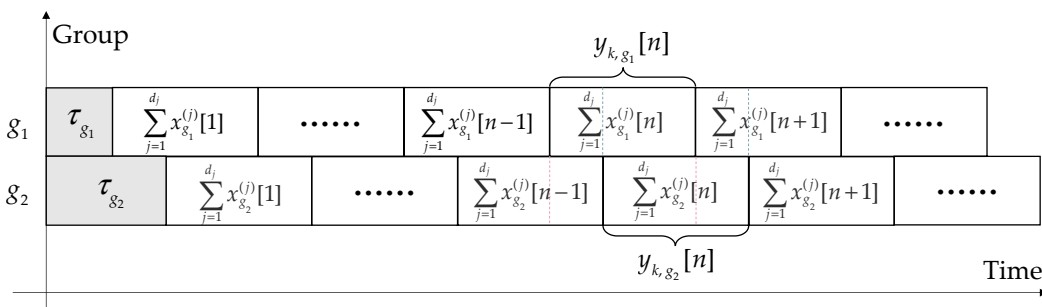

**Figure 7.** Receiving signal structure.

$n$ means the $n^{th}$ symbol, $\psi$ represents the subscript difference of adjacent symbols, which is 0 (current symbol), $-1$ (last symbol) or 1 (next symbol). $\xi_{g1}^{\psi}$ and $\xi_{g2}^{\psi}$ are functions of $\rho$ and $\psi$. $h_{k,g_1}^{(d_j)}$ and $h_{k,g_2}^{(d_j)}$ represent the Rayleigh channel gain of user $d_j$ on the $k^{th}$ resource block in group 1 and group 2. $x_{g_1}^{(d_j)}$ and $x_{g_2}^{(d_j)}$ mean the codeword sent by user $d_j$ and $w_k$ is the additive white Gaussian noise (AWGN), $w_k \sim \mathcal{CN}(0, \sigma^2)$.

## 3. Asynchronous Detection Algorithm

This section describes two methods to detect information of users in two groups jointly. First, the traditional MPA method is revised and represented as the asynchronous scheme. Then, the SP-MPA method is proposed to improve the system performance.

### 3.1. MIMO-SCMA Scheme

Because the multi-user grouping model increases the number of accessed users, the number of interference terms will also increase, resulting in a high bit error rate in the system. In order to solve this problem, the multi-antenna-diversity effect can be used to increase transmission reliability and accuracy.

In the multi-antenna system, each antenna in the base station receives the same superimposed signal from the users. The joint factor graph is constructed, as in Figure 8. Each antenna can be converted into a virtual function node [27] and there are $R \times K$ function nodes in one group, which corresponded to $R \times K$ rows. Each resource block accommodates $2 \times J$ users, which corresponded to $2 \times J$ columns. The corresponding joint factor matrix is in Figure 9. It can be seen from the factor graph that the amount of information propagation increases as the number of antennas increases, which can improve system performance.

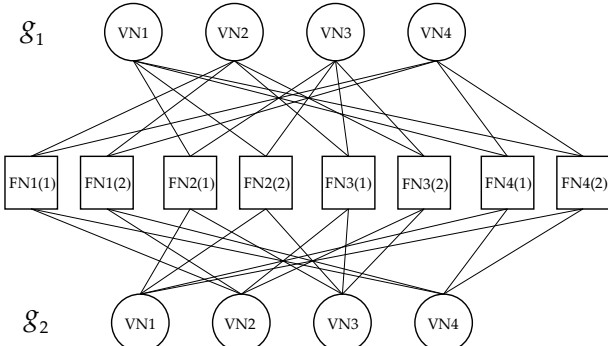

**Figure 8.** Joint factor graph.

$$f = \begin{bmatrix} 0 & 1 & 0 & 1 & 0 & 1 & 0 & 1 \\ 1 & 0 & 1 & 0 & 1 & 0 & 1 & 0 \\ 0 & 1 & 1 & 0 & 0 & 1 & 1 & 0 \\ 1 & 0 & 0 & 1 & 1 & 0 & 0 & 1 \\ 0 & 1 & 0 & 1 & 0 & 1 & 0 & 1 \\ 1 & 0 & 1 & 0 & 1 & 0 & 1 & 0 \\ 0 & 1 & 1 & 0 & 0 & 1 & 1 & 0 \\ 1 & 0 & 0 & 1 & 1 & 0 & 0 & 1 \end{bmatrix}$$

**Figure 9.** Joint factor matrix.

### 3.2. Asynchronous Log-MPA Method

MPA is an iterative algorithm based on factor graph. Through the calculation and propagation of edge probability between resource nodes and user nodes, the probability of each codeword is finally obtained. The final estimated bit stream is obtained by log likelihood ratio (LLR) decision.

In the synchronous detection method, each user obtains the codeword probability of each symbol through message transmission and the detection of each symbol is independent from each other. The traditional synchronization model can obtain more accurate results through the MPA algorithm. The model in this paper can also be detected by MPA. In order to reduce the exponential operation, log-MPA is used here.

First, the user codeword probability is initialized and then the message is passing according to the factor graph in Figure 8. Group 1 and Group 2 use the same factor graph,

where FN refers to the frequency resource node and VN is the user node. The process of asynchronous log-MPA is illustrated as follows.

$$M_{j\to k}^0(x^{(j)}) = \mathrm{P}(x^{(j)}) = \log(\frac{1}{M}), j = 1, \cdots, J, \forall k \in H_j \tag{6}$$

$$
\begin{aligned}
M_{k\to j}^t(x^{(j)}) &= \overset{*}{\max_{\{x^{(i)}|i\in\mathrm{T}_k\backslash j\}}} \left\{ -\frac{1}{\sigma^2} \|y_{k,g}[n] - \sum_{j\in\mathrm{T}_k} h_{k,g}^{(j)}[n] \times x_g^{(j)}[n] \right. \\
&\quad - \sum_{g\neq g}\sum_{\psi=0,\pm1} \xi_g^\psi \times \sum_{j\in\mathrm{T}_k} h_{k,g}^{(j)}[n+\psi] \times x_g^{(j)}[n+\psi]\|^2 \\
&\quad \left. + \sum_{i\in\mathrm{T}_k\backslash j} M_{i\to k}^{t-1}(x^{(i)}) \right\}
\end{aligned}
\tag{7}
$$

$$M_{j\to k}^t(x^{(j)}) = \log(\frac{1}{M}) + \sum_{i\in\mathrm{H}_j\backslash k} M_{i\to j}^{t-1}(x^{(i)}) - \sum_{x^{(i)}}\sum_{i\in\mathrm{H}_j\backslash k} M_{i\to j}^{t-1}(x^{(i)}) \tag{8}$$

$$
\begin{aligned}
\log(\exp(a_1 + \cdots + a_n)) &= \overset{*}{\max}(a_1, \cdots, a_n) = \max(a_1, \cdots, a_n) \\
&\quad + \log(1 + \sum_{i\in\{1,\cdots,n\}\backslash j} \exp(-|\max(a_1, \cdots, a_n) - a_i|)
\end{aligned}
\tag{9}
$$

In the formulas, $\mathrm{T}_k$ means the user set on the $k^{th}$ resource block and $\mathrm{H}_j$ means the resource set where the $j^{th}$ user occupies. $M_{k\to j}^t(x^{(j)})$ means the information passing from the $k^{th}$ resource block to the $j^{th}$ user node and $M_{j\to k}^t(x^{(j)})$ means the information passing from the $j^{th}$ user node to the $k^{th}$ resource block. The updating process of $M_{k\to j}^t(x^{(j)})$ is normalized and the definition of $\overset{*}{\max}$ is listed in (9) [10]. Then, the LLR is conducted to recover the bit stream of users. Since the decision process is the same for each user group, the codeword probability $Q(x^{(j)})$ and $LLR(b^{(j)}(i))$ of users in two groups are represented in the same way. The whole process is listed in Algorithm 1.

$$Q(x^{(j)}) = \log(\frac{1}{M}) + \sum_{i\in\mathrm{H}_j} M_{i\to j}^{t_{\max}}(x^{(j)}) \tag{10}$$

$$LLR\left(b^{(j)}(i)\right) = \log\left(\frac{\sum_{x^{(j)}|b^{(j)}(i)=0} \exp(Q(x^{(j)}))}{\sum_{x^{(j)}|b^{(j)}(i)=1} \exp(Q(x^{(j)}))}\right) \tag{11}$$

---

**Algorithm 1.** Asynchronous Log-MPA

---

| | |
|---|---|
| **Input:** | $y$, $h$, $\mathcal{C}_{g1}$, $\mathcal{C}_{g2}$, $\sigma^2$ |
| **Initialize:** | $\mathrm{P}(x^{(j)}) = \log(1/M)$, $M_{j\to k}^0(x^{(j)}) = \mathrm{P}(x^{(j)})$, $M_{k\to j}^0(x^{(j)}) = 0$ |
| 1: | **for** *n* = 1:N **do** |
| 2: | **for** *t* = 1:$t_{\max}$ **do** |
| 3: | **for** *k*= 1 :R × K **do** |
| 4: | use (7) to update $M_{k\to j}^t(x^{(j)})$ |
| 5: | **end for** |
| 6: | **for** *j*= 1 :J **do** |
| 7: | use (8) to update $M_{j\to k}^t(x^{(j)})$ |
| 8: | **end for** |
| 9: | **end for** |
| 10: | **for** *j*= 1 :J **do** |
| 11: | use (10) to calculate the probability of codewords |
| 12: | **end for** |
| 13: | use (11) to conduct LLR decision |
| 14: | **end for** |
| **Output:** | $b^{(j)}[n], n = 1 : N$ |

---

### 3.3. SP-MPA Method

In the above asynchronous Log-MPA method, each time the iteration is performed, the prior probability starts from $1/M$. The probability information of the preceding signal is not involved in the calculation; that is, the asynchronous Log-MPA does not make good use of the previous decision information. Therefore, a message-passing algorithm based on Serial Propagation (SP-MPA) is proposed to fully utilize the known information to improve the system performance. The whole process is listed in Algorithm 2.

The received signal is detected according to the asynchronous transmission structure and formula of the received signal. The asynchronous transmission process reveals that $y_{k,g_1}$ and $y_{k,g_2}$ contain common information, which can be detected, respectively, to obtain more accurate information values. First, we perform MPA on $y_{k,g_1}[n]$ and then perform MPA on $y_{k,g_2}[n]$. Because there exists the same symbol in $y_{k,g_1}[n]$ and $y_{k,g_2}[n]$, the decoded information can be utilized for each other. For example, the final probability of the first symbol of users in Group 1 in the detection of $y_{k,g_1}[1]$ can be used as the initial probability in the detection of $y_{k,g_2}[1]$.

Messages are passed between nodes and the node graph of the propagation process is shown in Figure 10. For clarity, only one user node shared on the first resource block is listed in the figure because each user on the same resource block has the same information updating mode. $F_g$ represents the likelihood probability of group $g$ observed on the $k^{th}$ resource block. $Q1$ and $Q2$ represent the output probability of codewords in Group 1 and Group 2 at the end of the iteration.

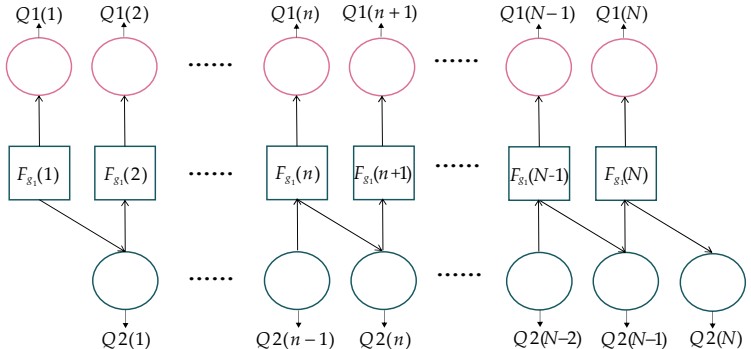

**Figure 10.** Node graph of the propagation process in SP-MPA.

The propagation formulas are listed as follows. When detecting the first symbol, the initial probability has no reference, so that its value equals $1/M$. Then, the initial probability can be obtained from the last detection, which is propagated forward.

$$M^0_{j \to k}(x^{(j)}_{g_1}[1]) = P(x^{(j)}_{g_1}[1]) = \log(\frac{1}{M}), j = 1, \cdots, J, \forall k \in H_j \tag{12}$$

$$M^0_{j \to k}(x^{(j)}_{g_2}[1]) = P(x^{(j)}_{g_2}[1]) = \log(\frac{1}{M}), j = 1, \cdots, J, \forall k \in H_j \tag{13}$$

$$M^0_{j \to k, y_{g1}[n]}(x^{(j)}_{g_1}[n]) = M^{t_{\max}}_{j \to k, y_{g2}[n-1]}(x^{(j)}_{g_1}[n]), n = 2, 3, \cdots N \tag{14}$$

$$M^0_{j \to k, y_{g2}[n]}(x^{(j)}_{g_2}[n]) = M^{t_{\max}}_{j \to k, y_{g1}[n]}(x^{(j)}_{g_2}[n]), n = 2, 3, \cdots N \tag{15}$$

$$
\begin{aligned}
M^t_{k \to j}(x^{(j)}) = \ \overset{*}{\underset{\{x^{(i)}|i \in T_k \backslash j\}}{\max}} &\left\{ -\frac{1}{\sigma^2} \Big\| y_{k,g}[n] - \sum_{j \in T_k} h^{(j)}_{k,g}[n] \times x^{(j)}_g[n] \right. \\
&\left. - \sum_{g \neq g} \sum_{\psi=0,\pm 1} \xi^\psi_g \times \sum_{j \in T_k} h^{(j)}_{k,g}[n+\psi] \times x^{(j)}_g[n+\psi] \Big\|^2 \right. \\
&\left. + \sum_{i \in T_k \backslash j} M^{t-1}_{i \to k}(x^{(i)}) \right\}
\end{aligned} \tag{16}
$$

$$M_{j \to k}^t(x^{(j)}) = \log(\frac{1}{M}) + \sum_{i \in H_j \backslash k} M_{i \to j}^{t-1}(x^{(i)}) - \sum_{x^{(i)}} \sum_{i \in H_j \backslash k} M_{i \to j}^{t-1}(x^{(i)}) \qquad (17)$$

Through the above algorithm, the previous information can be transmitted but the maximum number of iterations set by the traditional algorithm is not that accurate. Therefore, a precision is set to achieve a better convergence. If the difference between the codeword probability obtained by the previous iteration and the current iteration is smaller than the precision, the iteration is determined to be over. When the precision is bigger, it is easier for the difference in codewords to meet the precision and the iteration will end in advance, which reduces the complexity, but the accuracy will be reduced too.

| **Algorithm 2.** SP-MPA |
| --- |

| **Input:** | $y$, $h$, $C_{g1}$, $C_{g2}$, $\sigma^2$, $\varepsilon$ |
| --- | --- |
| **Initialize:** | $P(x^{(j)}) = \log(1/M)$, $M_{j \to k}^0(x^{(j)}) = P(x^{(j)})$, $M_{k \to j}^0(x^{(j)}) = 0$ |
| | conduct asynchronous Log-MPA to obtain the final probability of code-words |
| 1: | **for** $n = 2 : N$ **do** |
| 2: | **while** $\left| M_{j \to k}^t(x^{(j)}) - M_{j \to k}^{t-1}(x^{(j)}) \right| < \varepsilon$ |
| 3: | use (14) and (15) to obtain the initial probability |
| 4: | **for** $k = 1 : R \times K$ **do** |
| 5: | use (16) to update $M_{k \to j}^t(x^{(j)})$ |
| 6: | **end for** |
| 7: | **for** $j = 1 : J$ **do** |
| 8: | use (17) to update $M_{j \to k}^t(x^{(j)})$ |
| 9: | **end for** |
| 10: | **end while** |
| 11: | **for** $j = 1 : J$ **do** |
| 12: | use (10) to calculate the probability of codewords |
| 13: | **end for** |
| 14: | use (11) to conduct LLR decision |
| 15: | **end for** |
| **Output:** | $b^{(j)}[n]$, $n = 1 : N$ |

## 4. Partial Gaussian Approximation with Imperfect Synchronization in Group

The previous section described the asynchronous detection algorithms based on the model in Section 2. The delays in one group are assumed to be almost the same, which is the perfect synchronization. However, this hypothesis is ideal, which is hard to realize. Although we put the users with similar delays in one group, there still exists a difference between these users and the synchronization in one group is imperfect. The transmission process should be revised to Figure 11.

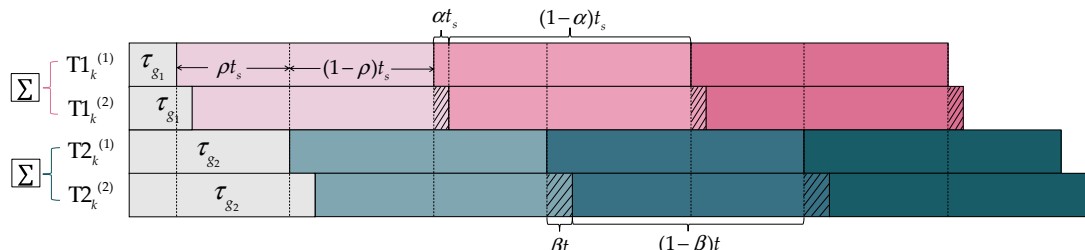

**Figure 11.** Asynchronous transmission process with imperfect synchronization in group.

### 4.1. Imperfect Scheme

In Figure 11, the delays of users in the same set have a slight gap: $\alpha t_s$ in $T1_k$ and $\beta t_s$ in $T2_k$, remarked with oblique lines, which will affect the detection of subsequent symbols. For example, the second symbol of $T1_k^{(1)}$ and the first symbol of $T2_k^{(1)}$ and $T2_k^{(2)}$ will be interfered by the first symbol of $T1_k^{(2)}$. Given that the users in the same group have similar delays, $\alpha$ and $\beta$ are much smaller than $\rho$. Figure 12 illustrates the constellation distances

of the two groups. For simplicity, only one point in Group 1 is drawn and the average distance between points in Group 1 and Group 2 is described as $1 - \rho$. In addition, the value of $h_{k,g}^{(d_j)}[1]$ is considered similar in one group to show constellations clearly. The pink squares are the first superimposed symbol of group 1 which should be detected, the green circles are the first superimposed symbol of group 2 which should be detected and the green squares are the final superimposed symbols of all users.

$$y_{k,g_1}[1] = \sum_{j=1}^{d_j} h_{k,g_1}^{(1)}[1] \times x_{g_1}^{(j)}[1] + (1 - \rho) \times \sum_{j=1}^{d_j} h_{k,g_1}^{(1)}[1] \times x_{g_1}^{(j)}[1] + w_k[1] \tag{18}$$

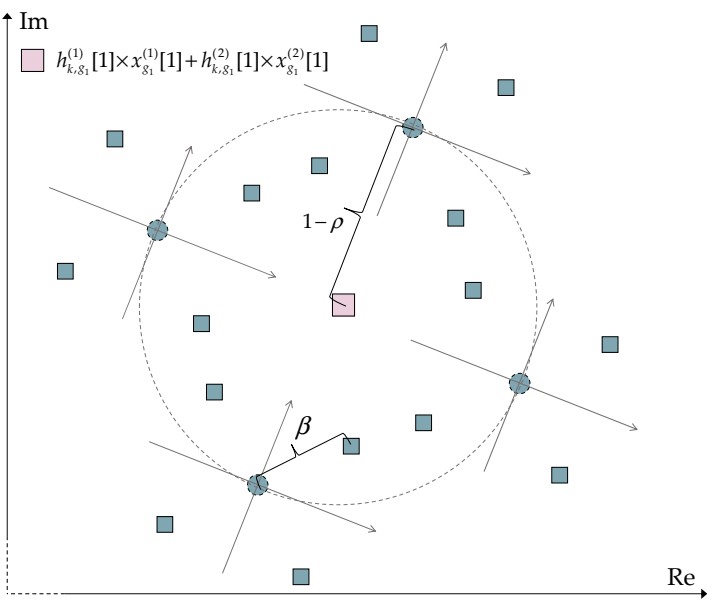

**Figure 12.** Constellation points of the first symbol.

If the algorithms based on perfect asynchronization in a group are applied in this model, the symbols delayed will not be taken into consideration, causing a big performance loss. However, if all symbols are added into the iteration, the computing complexity will be raised to a higher level. Therefore, new algorithms should be proposed to balance the performance and the complexity.

$$
\begin{aligned}
y_{k,g}[n] &= \left\{ h_{k,g}^{(1)}[n] \times x_g^{(1)}[n] + \sum_{\psi_1 = 0, \pm 1} \chi_g^{\psi_1} h_{k,g}^{(2)}[n + \psi_1] \times x_g^{(2)}[n + \psi_1] \right\} \\
&+ \sum_{g \neq g} \left\{ \sum_{\psi = 0, \pm 1} \chi_g^{\psi} \times h_{k,g}^{(1)}[n + \psi] \times x_g^{(1)}[n + \psi] + \sum_{\psi_2 = 0, \pm 1} \chi_g^{\psi_2} \times h_{k,g}^{(2)}[n + \psi_2] \times x_g^{(2)}[n + \psi_2] \right\} + w_k[n]
\end{aligned} \tag{19}
$$

$$
\chi_{g_1}^{\psi_1} = \begin{cases} 1 - \alpha & , \psi_1 = 0 \\ \alpha & , \psi_1 = -1 \\ 0 & , \psi_1 = 1 \end{cases} \quad
\chi_{g_2}^{\psi} = \begin{cases} 1 - \rho & , \psi = 0 \\ \rho & , \psi = -1 \\ 0 & , \psi = 1 \end{cases} \quad
\chi_{g_2}^{\psi_2} = \begin{cases} 1 - \rho - \beta & , \psi_2 = 0 \\ \rho + \beta & , \psi_2 = -1 \\ 0 & , \psi_2 = 1 \end{cases} \tag{20}
$$

$$
\chi_{g_2}^{\psi_1} = \begin{cases} 1 - \beta & , \psi_2 = 0 \\ \beta & , \psi_2 = -1 \\ 0 & , \psi_2 = 1 \end{cases} \quad
\chi_{g_1}^{\psi} = \begin{cases} 1 - \rho & , \psi = 0 \\ 0 & , \psi = -1 \\ \rho & , \psi = 1 \end{cases} \quad
\chi_{g_1}^{\psi_2} = \begin{cases} 1 - \rho + \alpha & , \psi_2 = 0 \\ 0 & , \psi_2 = -1 \\ \rho - \alpha & , \psi_2 = 1 \end{cases} \tag{21}
$$

$$
\begin{aligned}
y_{k,g_1}[n] &= \left\{ h_{k,g_1}^{(1)}[n] \times x_{g_1}^{(1)}[n] + \sum_{\psi_1 = 0,+1} \chi_{g_1}^{\psi_1} h_{k,g_1}^{(2)}[n + \psi_1] \times x_{g_1}^{(2)}[n + \psi_1] \right\} \\
&+ \sum_{g \neq g_1} \left\{ \sum_{\psi = 0,\pm 1} \chi_g^{\psi} \times h_{k,g}^{(1)}[n + \psi] \times x_g^{(1)}[n + \psi] + \sum_{\psi_2 = 0,\pm 1} \chi_g^{\psi_2} \times h_{k,g}^{(2)}[n + \psi_2] \times x_g^{(2)}[n + \psi_2] \right\} \\
&+ \underbrace{\left\{ \chi_{g_1}^{-1} h_{k,g_1}^{(2)}[n-1] \times x_{g_1}^{(2)}[n-1] + w_k[n] \right\}}_{\triangleq z_{k,g_1}[n]}
\end{aligned}
\tag{22}
$$

Consider the received signal in symbol-related form in (3). The whole formula should be revised and written as (19). $\psi$ means the difference in subscript difference of adjacent symbols. There are three kinds of delays: the first user in $T1_k$ and $T2_k$, the second user in $T1_k$ and the second user in $T2_k$, so there are three kinds of $\chi_g^{\psi}$: $\chi_g^{\psi_1}$, $\chi_g^{\psi_2}$ and $\chi_g^{\psi}$.

Further, the formula can be represented with signals needed and signals that are considered as Gaussian noise. Take $y_{k,g_1}[n]$, for example. In $y_{k,g_1}[n]$, the last symbol of user 2 in Group 1 is the interference, which should be separated as the Gaussian noise. The last item in (22) includes the original Gaussian noise and the product of the last symbol of user 2 in Group 1 and its corresponding channel gain [28]. The formula of $y_{k,g_2}[n]$ is derived in the same way, so only the process of $y_{k,g_1}[n]$ is outlined.

$z_{k,g_1}[n]$ is modeled as $z_{k,g_1}[n] : \mathcal{CN}(\mu_k[n], \sigma^2_k[n])$ where $\mu_k[n]$ and $\sigma^2_k[n]$ are the mean and variance of $z_{k,g_1}[n]$, respectively, which are represented as follows. $\mathcal{C}_{g_1,2}(k,m)$ means the $m^{th}$ codeword of user 2 on the $k^{th}$ resource block in group 1. $M_{2 \to k}^{t_{\max}}(x^{(2)} = m)$ means the probability of the $m^{th}$ codeword of user 2 on the $k^{th}$ resource block in Group 1, which is obtained from the last symbol detection.

Therefore, the message passing from the function node to the user node can be written as (25) and (26). The codeword probability $Q(x^{(j)})$ and $LLR(b^{(j)}(i))$ are the same as (10) and (11).

$$
\mu_k[n] = h_{k,g_1}^{(2)}[n-1] \sum_{m=1}^{M} \mathcal{C}_{g_1,2}(k,m) M_{2 \to k}^{last}(x^{(2)} = m)
\tag{23}
$$

$$
\begin{aligned}
\sigma^2_k[n] &= h_{k,g_1}^{(2)}[n-1]\big|_2 + \\
&\left\{ \sum_{m=1}^{M} \left| \mathcal{C}_{g_1,2}(k,m) \right|_2 M_{2 \to k}^{last}(x^{(2)} = m) - \left| \mu_k[n] \right|^2 \right\} + \sigma^2
\end{aligned}
\tag{24}
$$

$$
\begin{aligned}
M_{k \to j}^{t}(x^{(j)}) &= \overset{*}{\max_{\{x^{(i)} | i \in T_k \backslash j\}}} \left\{ -\frac{1}{\sigma^2_k[n]} \| y_{k,g}[n] - \left\{ h_{k,g_1}^{(1)}[n] \times x_{g_1}^{(1)}[n] + \sum_{\psi_1 = 0,+1} \chi_{g_1}^{\psi_1} h_{k,g_1}^{(2)}[n + \psi_1] \times x_{g_1}^{(2)}[n + \psi_1] \right\} \right. \\
&\left. - \sum_{g \neq g_1} \left\{ \sum_{\psi = 0,\pm 1} \chi_g^{\psi} \times h_{k,g}^{(1)}[n + \psi] \times x_g^{(1)}[n + \psi] + \sum_{\psi_2 = 0,\pm 1} \chi_g^{\psi_2} \times h_{k,g}^{(2)}[n + \psi_2] \times x_g^{(2)}[n + \psi_2] \right\} - \mu_k[n] \|^2 \right. \\
&\left. + \sum_{i \in T_k \backslash j} M_{i \to k}^{t-1}(x^{(i)}) \right\}
\end{aligned}
\tag{25}
$$

$$
M_{j \to k}^{t}(x^{(j)}) = \log\left(\frac{1}{M}\right) + \sum_{i \in H_j \backslash k} M_{i \to j}^{t-1}(x^{(i)}) - \sum_{x^{(i)}} \sum_{i \in H_j \backslash k} M_{i \to j}^{t-1}(x^{(i)})
\tag{26}
$$

However, if the Gaussian approximation (GA) method is conducted on each resource block, the accuracy of detection will experience large degradation. To balance the detection performance and complexity, a partial Gaussian approximation (PGA) method [29] will be applied, in which only parts of resource blocks adopt the GA method with other blocks using Log-MPA. In Figure 13, for example, three resource blocks adopt PGA, which are marked with blue lines.

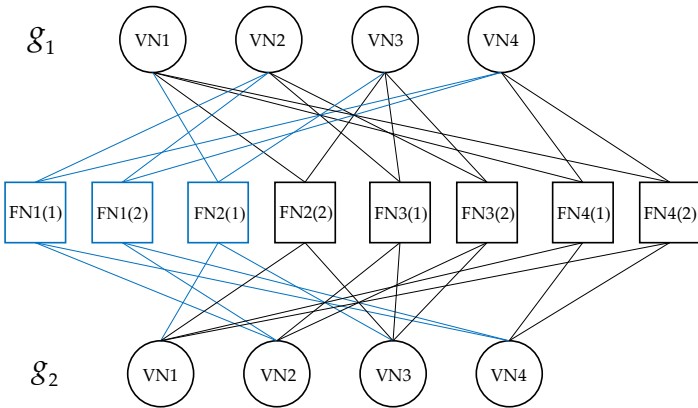

**Figure 13.** Joint-factor graph with PGA.

From the above PGA process, the imperfect synchronization in the group can be solved by regarding the interference as the Gaussian noise on resource blocks. At the same time, the complexity is reduced because the delayed symbol in one group is not computed in the iteration. The whole algorithm is listed in Algorithm 3.

| **Algorithm 3.** PGA-MPA |
|---|
| **Input:** $\quad y, h, \mathcal{C}_{g1}, \mathcal{C}_{g2}, \sigma^2, \varepsilon$ |
| **Initialize:** $\quad \mathrm{P}(x^{(j)}) = \log(1/M), M^0_{j\to k}(x^{(j)}) = \mathrm{P}(x^{(j)}), M^0_{k\to j}(x^{(j)}) = 0$ |
| $\qquad\qquad$ conduct asynchronous Log-MPA to obtain the final probability of code-words |
| 1: $\qquad$ **for** $n= 2 :N$ **do** |
| 2: $\qquad\quad$ **while** $\left| M^t_{j\to k}(x^{(j)}) - M^{t-1}_{j\to k}(x^{(j)}) \right| < \varepsilon$ |
| 3: $\qquad\qquad$ use (14) and (15) to obtain the initial probability |
| 4: $\qquad\qquad$ use (23) and (24) to compute $\mu_k[n]$ and $\sigma^2_k[n]$ |
| 5: $\qquad\qquad$ **for** $k= 1 :K_1$ **do** |
| 6: $\qquad\qquad\quad$ use (25) to update $M^t_{k\to j}(x^{(j)})$ |
| 7: $\qquad\qquad$ **end for** |
| 8: $\qquad\qquad$ **for** $j= 1 :J$ **do** |
| 9: $\qquad\qquad\quad$ use (26) to update $M^t_{j\to k}(x^{(j)})$ |
| 10: $\qquad\qquad$ **end for** |
| 11: $\qquad\quad$ **end while** |
| 12: $\qquad\quad$ **for** $j= 1 :J$ **do** |
| 13: $\qquad\qquad$ use (10) to calculate the probability of codewords |
| 14: $\qquad\quad$ **end for** |
| 15: $\qquad\quad$ use (11) to conduct LLR decision |
| 16: $\qquad\quad$ **end for** |
| **Output:** $\quad b^{(j)}[n], n = 1 : N$ |

### 4.2. Complexity Analysis

The complexity of the three algorithms mentioned before is listed in Table 1. Log-MPA and SP-MPA almost share the same complexity in the perfect scheme; the only difference is the number of iterations. From the simulation, the convergence ability is almost the same so the number of iterations is almost the same too.

In the imperfect scheme, SP-MPA (interference ignored) (SP-MPA (II)) has the lowest complexity, for it does not consider the interference and SP-MPA has the highest complexity, for it considers all symbols into iterations. In PGA-MPA, the complexity is reduced from $M^{3\times d_j+1} \times R \times K$ to $M^{3\times d_j+1} \times K_1 + M^{3\times d_j} \times (R \times K - K_1)$ exponentially.

**Table 1.** Computing complexity in different schemes.

| Algorithm | Scheme | Complexity |
|-----------|--------|------------|
| Log-MPA | Perfect scheme | $[(3 \times d_j - 1) \times M^{3 \times d_j} \times (3 \times d_j) \times R \times K + d_k \times M \times J] \times t_{\max}$ |
| SP-MPA | Perfect scheme | $[(3 \times d_j - 1) \times M^{3 \times d_j} \times (3 \times d_j) \times R \times K + d_k \times M \times J] \times t_{final}$ |
| SP-MPA (II) | Imperfect scheme | $[(3 \times d_j - 1) \times M^{3 \times d_j} \times (3 \times d_j) \times R \times K + d_k \times M \times J] \times t_{\max}$ |
| SP-MPA | Imperfect scheme | $[(3 \times d_j) \times M^{3 \times d_j + 1} \times (3 \times d_j + 1) \times R \times K + d_k \times M \times J] \times t_{final}$ |
| PGA-MPA | Imperfect scheme | $[(3 \times d_j) \times M^{3 \times d_j + 1} \times (3 \times d_j + 1) \times K_1 + (3 \times d_j - 1) \times M^{3 \times d_j}$ $\times 3 \times d_j \times (R \times K - K_1) + d_k \times M \times J] \times t_{final}$ |

In the perfect scheme, $d_j$ is smaller than that in the imperfect scheme, so the computing operations of these algorithms are also fewer than that in the imperfect scheme.

## 5. Simulation Results

In this section, simulation and numerical analysis are conducted in different schemes. The superiority of SP-MPA and PGA-MPA is verified in the grouped system. The parameters in this paper are listed in Table 2. The transmitting signal power is normalized to 1 in this paper.

**Table 2.** Parameters and values in proposed algorithms.

| Parameter | Meaning | Value |
|-----------|---------|-------|
| $K$ | Number of resource blocks | 4 |
| $J$ | Number of users in one group | 4 |
| $R$ | Number of antennas of the base station | 1, 2, 4 |
| $d_j$ | Number of users on the same resource block | 2 |
| $d_k$ | Number of resource blocks occupied by the same user | 2 |
| $M$ | The modulation order (QPSK) | 4 |
| $d_{g_1}$ | The distance from the near group to the base station | 0.5 (km) |
| $d_{g_2}$ | The distance from the far group to the base station | 1.0 (km) |
| $N$ | Number of transmitted symbols | 20,000 |
| $t_{\max}$ | The maximum number of iterations | 6 |
| $\alpha$ | delay gap in group 1 | [0.1–0.3] |
| $\beta$ | delay gap in group 2 | [0.1–0.3] |
| $\rho$ | delay gap between different groups | [0.3–0.7] |
| $P_{g_1}$ | Normalized transmitting power of users in group 1 | 1 |
| $P_{g_2}$ | Normalized transmitting power of users in group 2 | 0.8 |

Figure 14 shows the system-average BER performance of Log-MPA and SP-MPA, in which $g_1$ represents the near-user group and $g_2$ represents the far-user group. Using SP-MPA can greatly reduce the bit error rate and improve the system performance, especially under the condition of high signal-to-noise ratio. It can be seen from the figure that the downward trend of Log-MPA is not obvious with the bit error rate being at a high level, while the curve of SP- MPA begins to decline rapidly when Eb/N0 is greater than 15 dB and the bit error rate is close to $10^{-5}$ when Eb/N0 is 30 dB. When Eb/N0 is smaller (less than 15 dB), the incorrect probability of codewords in the previous detection will affect the next symbols; thus, the curve of SP-MPA is closer to that of log-MPA. When Eb/N0 is higher, the correct ability will become higher; thus, the gap between these two algorithms will become larger.

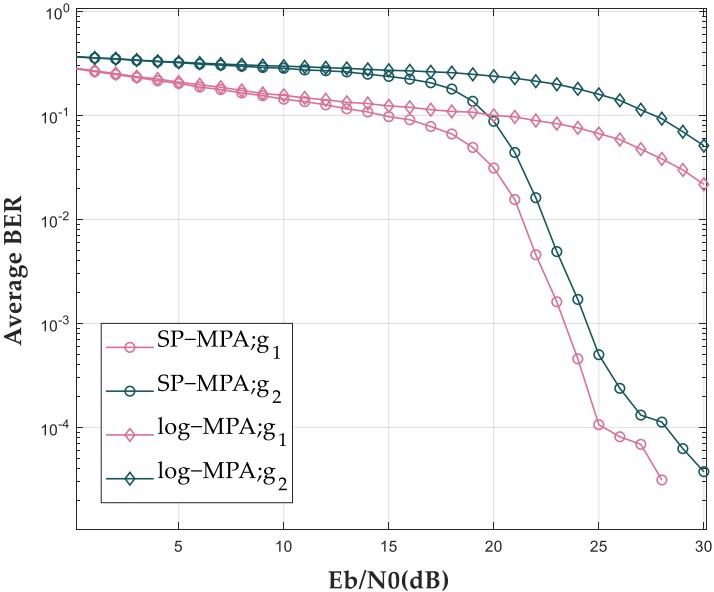

**Figure 14.** Average BER curves of log−MPA and SP−MPA.

The reason is that the initial codeword probability of Log-MPA starts with $1/M$ at each iteration, while SP-MPA retains the probability of the previous detection as the initial probability. Previous codeword probability has been fully calculated so it is not necessary to calculate from scratch in this detection.

Figure 15 illustrates the results with different numbers of users in two groups. When user activity is higher, more users share the same resource and the average BER would be increased quickly.

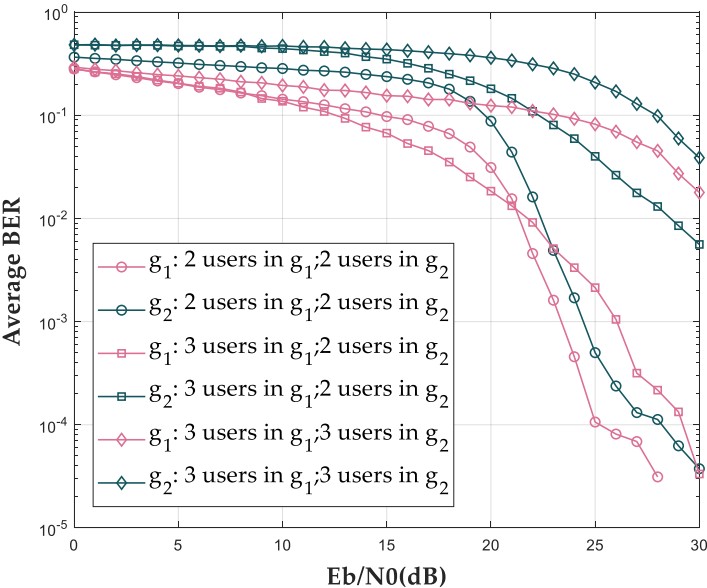

**Figure 15.** Average BER of two groups with different numbers of users.

Figure 16 shows the bit error rate performance of the MIMO system with different numbers of receiving antennas at the base station. Increasing the number of receiving antennas is equivalent to increasing the number of virtual resource nodes. When the number of transmission paths increases, the reliability of code probability is improved. When Eb/N0 is 15 dB, the average bit error rate in Group 1 and Group 2 is 0.1079 and 0.2487, respectively, when $R$ is 1. The average bit error rate is 0.0167 and 0.0873 when $R$ is 2 and

0.0007 and 0.0141 when *R* is 4. It can be seen that the diversity effect of multiple antennas brings lower bit error rate and more reliable performance.

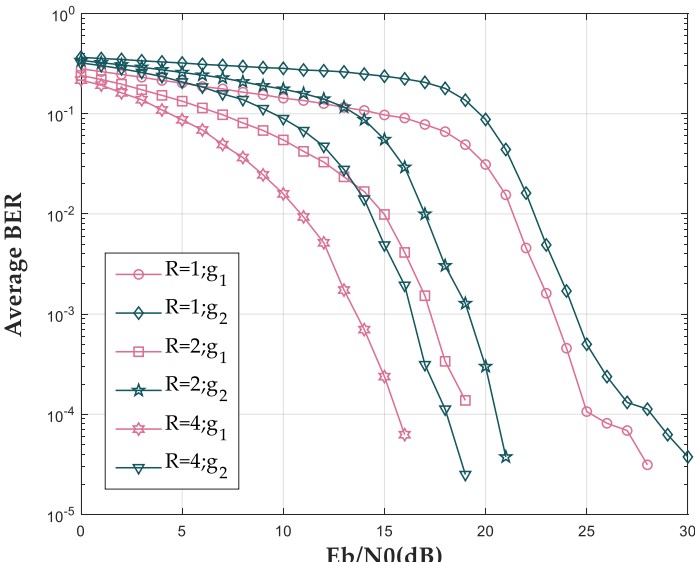

**Figure 16.** Average BER curves of SP−MPA in MIMO system.

In the imperfect scheme, if the interference is ignored (SP-MPA (interference ignored), SP-MPA (II)), the decoding performance is too bad to obtain the correct signals sent by users. It can be seen from Figure 17 that the SP-MPA (II) curve is almost a straight line without falling down. PGA-MPA computes the mean value and variance in the delayed symbol, so the correctability is higher than SP-MPA (II). Its correctability is lower than SP-MPA, owing to the delayed symbol is not involved in the iterations but only taken as Gaussian noise.

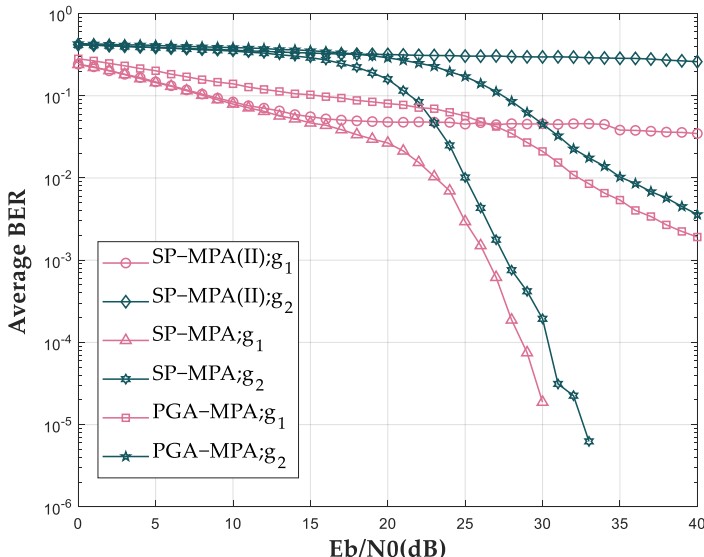

**Figure 17.** Average BER curves in imperfect scheme.

The antennas can be equalized to virtual function nodes, which can perform PGA-MPA. In Figure 18, when $K_1$ increases, more blocks adopt GA and the BER increases because of the approximation. However, the complexity also drops because of the reduction in nodes involved in the iterations.

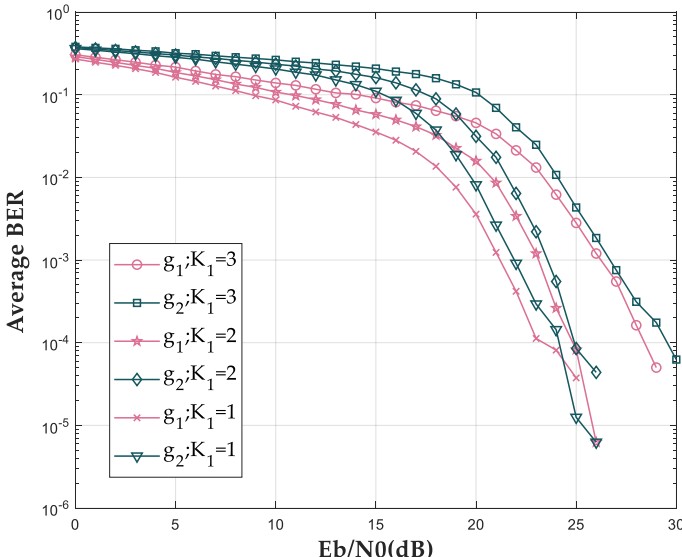

**Figure 18.** Impact of different $K_1$ in PGA−MPA.

Figure 19 shows the complexity and bit error rate of each algorithm in different schemes and the specific values are listed in Table 3. In the perfect scheme, Log-MPA and SP-MPA have similar complexity. In the simulation of convergence in Figure 20, the iterations are almost the same, which means the complexity is decided mainly by the iterations inside. The values on the y-axis in Figure 20 are the posterior probability of codewords (QPSK modulated symbols) in log format during the soft decoding processing; thus, higher value leads to higher detection probability. Compared to SP-MPA, the complexity of PM-MPA in [7] is lower but the bit error rate is much higher, reducing the reliability of communication. In the imperfect scheme, SP-MPA(II) has the lowest complexity with the worst performance and SP-MPA has the highest complexity with the best performance. PGA-MPA decreases some exponential operations and it balances the complexity and performance. In Figure 19, $K_1$ is set as 1. In order to eliminate unrelated factors, the operations and time are normalized.

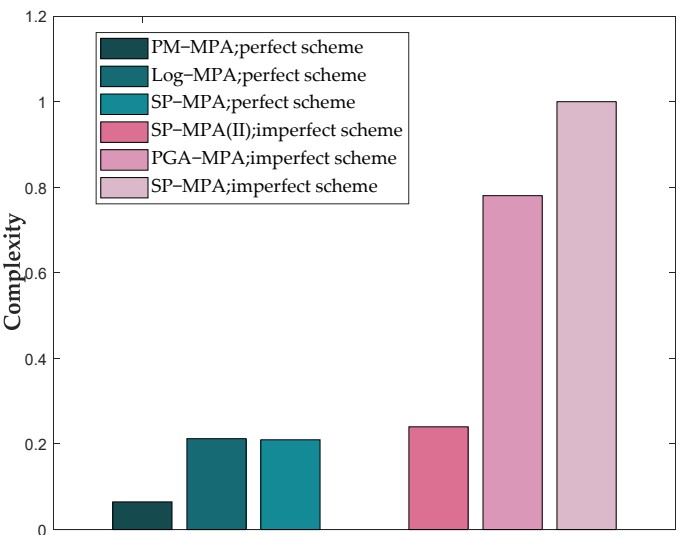

**Figure 19.** Convergence of the codeword probability.

**Table 3.** Complexity and bit error rate in different algorithms.

| Algorithm | Scheme | Complexity (Normalized) | Bit Error Rate (30 dB) |
|---|---|---|---|
| Log-MPA | Perfect | 0.2122 | $2.1713 \times 10^{-2}$ |
| SP-MPA | Perfect | 0.2097 | $2.0167 \times 10^{-5}$ |
| PM-MPA | Perfect | 0.0644 | $6.6250 \times 10^{-2}$ |
| SP-MPA(II) | Imperfect | 0.2401 | $4.5256 \times 10^{-2}$ |
| SP-MPA | Imperfect | 1.0000 | $1.8750 \times 10^{-5}$ |
| PGA-MPA | Imperfect | 0.7803 | $2.1056 \times 10^{-2}$ |

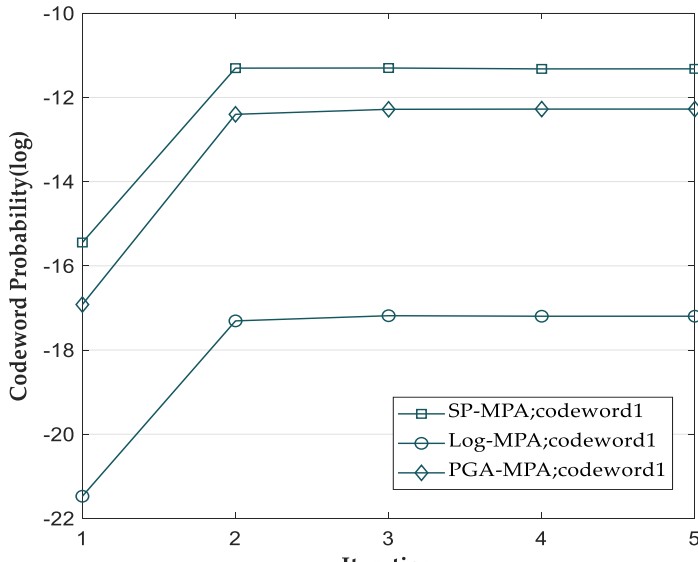

**Figure 20.** Normalized complexities of different algorithms.

Figure 21 shows the bit error rate with groups, which have different delay gaps $\alpha$ in one group. Here, only $\alpha$ is compared, for $\beta$ has the same impact. It can be seen that smaller $\alpha$ leads to a lower bit error rate because of the smaller interference of delayed symbol. Therefore, users with similar delays have to be grouped in the same collection.

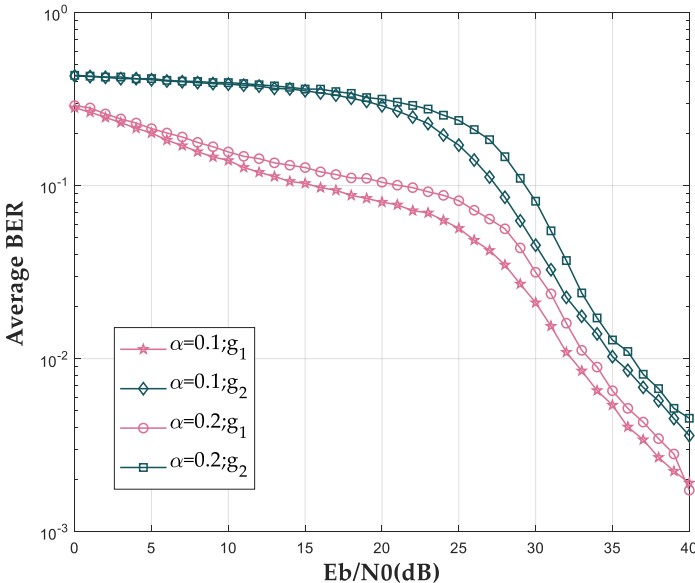

**Figure 21.** Impact of different $\alpha$ in PGA−MPA.

## 6. Conclusions

In this paper, an asynchronous grouped MIMO-SCMA system model is presented and its related detection algorithms are studied. First, with a perfect case in one group, the SP-MPA is proposed to utilize the information of previous detection and reduce the bit error rate of the system. Secondly, to compensate the imperfect synchronization in one group, the PGA-MPA is proposed to approximate the interference on parts of the resource blocks, which balances the complexity and correct ability, achieving a better detecting result. However, only four transmit antennas are considered in this paper due to the complexity problem. The cases with more antennas should be studied further.

**Author Contributions:** Conceptualization and methodology, X.W., N.Z. and J.H.; software and validation, X.W., N.Z.; writing—original draft preparation, X.W.; writing—review and editing, X.W.; supervision and project administration, J.H. All authors have read and agreed to the published version of the manuscript.

**Funding:** This research was funded by National Natural Science Foundation of China, grant number 62071319.

**Data Availability Statement:** Not applicable.

**Conflicts of Interest:** The authors declare no conflict of interest.

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
