# Peer review of "Asynchronous Partial Gaussian Approximation Detection Algorithm for Uplink-Grouped MIMO-SCMA System"

_electronics, doi:10.3390/electronics11213536_

Round 1

Reviewer 1 Report

Authors have presented an asynchronous partial gaussian approximation detection algorithm for uplink grouped MIMO-SCMA system. The paper is well written and the results are well explained. However, the following modifications should be corrected in the manuscript:

-        Kindly improve the abstract and add some advantages and quantitative results for the proposed method.

-        How are the parameters and values are considered in Table 2. Are the effects of all these parameters considered on the performance of the proposed algorithm?

-        Kindly compare the obtained results of the proposed algorithm with the related works, such as [14-18].

-        Could you explain about the effects of the user activity?

-        Also, the effects of different SNRs are not considered in the manuscript. 

Author Response

The main revisions

  1. The abstract has been revised and some numerical results are added to prove the superiority of this paper. (Abstract and section 5)
  2. Some parameters which have impact on the system performance are added in Table 2. (section 5)
  3. The effect of user activity is explained in this paper. The results are illustrated in Figure 15. (section 5)
  4. Compared results of related works are shown in Table 3 and Figure 20. (section 5)
  5. Some mistakes are corrected: In Figure 14, the label of ‘’ has been revised to ‘’; In Figure 20, the label of PGA-MPA (imperfect scheme) and SP-MPA (imperfect scheme) are changed; In Figure 21, the values ofare changed to 0.1 and 0.2.(section 5)

6.The conclusion is revised and the prospection of further research is added. (conclusion)

Response to the reviewer 1’s comments

Comment 1: Kindly improve the abstract and add some advantages and quantitative results for the proposed method.

Response 1: Thank you for your constructive comments on my manuscript.

The abstract is revised according to your suggestion, in which some quantitative results are added to describe the advantages of the proposed methods, such as “With assuming the perfect synchronization in one group, the message passing algorithm based on serial propagation (SP-MPA) is proposed to reduce the bit error rate (BER), which could transfer the updated information to the next symbol as its initial probability. Furthermore, with more practical case, the partial gaussian approximation method (PGA) is designed to decrease the interference resulted from the imperfect synchronization in one group. As the result, the computing complexity of the proposed PGA method could be decreased at least 20% compared with SP-MPA and the BER could be improved about 10%”.  

Comment 2: How are the parameters and values considered in Table 2. Are the effects of all these parameters considered on the performance of the proposed algorithm?

Response 2: Thank you for your question.

Firstly, the parameters in the system model are considered according to the real communication environment and the references. For example, R is the antenna number of the base station and,are related to the SCMA codebook mapping. The values are mostly decided according to the references for comparison such as,are set as 4 and 6 because it’s a typical set of resources and users.    

Secondly, all the effects of these parameters in the table are considered in the performance of proposed algorithms. Some of them are fixed values, and the main parameters such as the number of antennas and delays are varied during possible ranges. 

Comment 3: Kindly compare the obtained results of the proposed algorithm with the related works, such as [14-18].

Response 3: Thank you for your constructive comments on my manuscript.

More simulation results are compared and listed in section 5 (Table 3 and Figure 20). Compared to SP-MPA, the complexity of PM-MPA in [7] is lower but the bit error rate is much higher, reducing the reliability of communication.  

Comment 4: Could you explain about the effects of the user activity?

Response 4: Thank you for your question.

More users in the group or in the resource block, the BER would be increased. Some related results are added in Figure 15. And the corresponding explanation is added too. Figure 15 illustrates the results with different number of users in two groups. When user activity is higher, more users share the same resource, and the average BER would be increased quickly.

Comment 5: Also, the effects of different SNRs are not considered in the manuscript.

Response 5: Thank you for your constructive comments on my manuscript.

The different SNRs are considered in this paper, and they are described as different Eb/N0 (bit level) in the figures, since the proposed decoding algorithms are in bit level. 

Reviewer 2 Report

1)   In Line 113, "The different delays will cause different cases in the detection of one symbol". Please reframe the sentence.

2)   In line 120, "Assuming that all users share the same environment around." please justify it.

3)   In line 149, "assuming that the delay of users in each user group to the base station is almost the same". Please justify as the distance is actually different.

4)   mMIMO technologies allow higher order MIMO configurations with > 64 antennas. Why authors have limited the Number of antennas at base station and user are limited to 4 and 1 respectively? 

5)   What is the transmitting signal power?

6)   There is grammatical mistake in line no. 387.

7)   From Fig. 14, Why Avg. BER is closely same if Eb/No < 15dB for SP-MPA and log-MPA techniques?

8)   If Fig. 18, what are the absolute values of code probabilities labelled on Y-axis?

9)   In Fig. 19, the proposed PGA-MPA method has highest complexity. you can please justify.

Author Response

The main revisions

  1. The abstract has been revised and some numerical results are added to prove the superiority of this paper. (abstract and section 5)
  2. Some parameters which have impact on the system performance are added in Table 2. (section 5)
  3. The effect of user activity is explained in this paper. The results are illustrated in Figure 15. (section 5)
  4. Compared results of related works are shown in Table 3 and Figure 20. (section 5)
  5. Some mistakes are corrected: In Figure 14, the label of ‘’ has been revised to ‘’; In Figure 20, the label of PGA-MPA (imperfect scheme) and SP-MPA (imperfect scheme) are changed; In Figure 21, the values ofare changed to 0.1 and 0.2.(section 5)

     6.The conclusion is revised and the prospection of further research is added. (conclusion)

Response to the reviewer 2’s comments

Comment 1: In Line 113, "The different delays will cause different cases in the detection of one symbol". Please reframe the sentence.

Response 1: Thank you for your constructive comments on my manuscript.

The sentence has been revised to ”As shown in Figure 1, different delays from the asynchronous multi-users would lead to various symbol detection cases”.

Comment 2: In line 120, "Assuming that all users share the same environment around." please justify it. 

Response 2: Thank you for your constructive comments on my manuscript.

The sentence is rewritten as “Assuming that all users are in the same environment but only with different distances to the base station in the system model, we can divide the users with similar distances into one group. For example, there are two groups in the system model. The users with the distances similar to the could be grouped as g1, and the users with the distances similar to the  could be grouped as g2.”  

Comment 3: In line 149, "assuming that the delay of users in each user group to the base station is almost the same". Please justify as the distance is actually different.

Response 3: Thank you for your constructive comments on my manuscript.

The distances of users in the system model are actually different. But users with similar distances to the base station are divided into one group, so that delays of the users in same group could be assumed as one same value if the perfect detection scheme is used.

Comment 4: MIMO technologies allow higher order MIMO configurations with > 64 antennas. Why authors have limited the Number of antennas at base station and user are limited to 4 and 1 respectively?

Response 4: Thank you for your question.

MIMO technologies could allow higher order MIMO configurations with > 64 antennas but also could cause higher computing complexity. As a simple example to explain the proposed algorithm, only 4 antennas are considered in this paper. And more antennas cases will be considered in the future research results.

Comment 5: What is the transmitting signal power?

Response 5: Thank you for your question.

The transmitting signal power is normalized to 1 in this paper. And all users are using the same transmitting power in the system model.  

Comment 6: There is grammatical mistake in line no. 387.

Response 6: Thank you for your constructive comments on my manuscript.

The original sentence is revised to ”Figure 14 shows the system performance of Log-MPA and SP-MPA, in which g1 represents the near user group and g2 represents the far user group”.

Comment 7: From Fig. 14, Why Avg. BER is closely same if Eb/N0 < 15dB for SP-MPA and log-MPA techniques?

Response 7: Thank you for your question.

SP-MPA is performed by transmitting the probability of codewords to the detection of next symbol, so the error will also be propagated to the next symbol. When Eb/N0 is smaller (less than 15dB), the incorrect probability of codewords in previous detection will affect the next symbols, thus the curve of SP-MPA is closer to that of log-MPA. When Eb/N0 gets larger, the correct ability will become higher, thus the gap between those algorithms will become larger. The explanation is added in the revised manuscript.

Comment 8: In Fig. 18, what are the absolute values of code probabilities labelled on Y-axis?

Response 8: Thank you for your question.

The values on y-axis in Figure 19 are the posterior probability of codewords (QPSK modulated symbols) in log format during the soft decoding processing, thus the higher value leads to higher detected probability.

Comment 9: In Fig. 19, the proposed PGA-MPA method has highest complexity. you can please justify.

Response 9: Thank you for your constructive comments on my manuscript.

I’m sorry that there is a mistake. The legends of PGA-MPA and SP-MPA were labelled adversely. Figure 20 has been revised.

Round 2

Reviewer 1 Report

All of the comments are considered in the manuscript and the paper can be accepted after correcting some typos and minor modifications.

-        Read the manuscript carefully and correct the typos. For example, ref [27] and [28] are not cited in the numerical order in the text.